# The impact of tocilizumab on respiratory support states transition and clinical outcomes in COVID-19 patients. A Markov model multi-state study

Jovana Milic [1‡]*, Federico Banchelli[2,3‡], Marianna Meschiari[4], Erica Franceschini[4], Giacomo Ciusa[4], Licia Gozzi[4], Sara Volpi[4], Matteo Faltoni[4], Giacomo Franceschi[4], Vittorio Iadisernia[4], Dina Yaacoub[4], Giovanni Dolci[4], Erica Bacca[4], Carlotta Rogati[4], Marco Tutone [4], Giulia Burastero[4], Alessandro Raimondi[4], Marianna Menozzi[4], Gianluca Cuomo[4], Luca Corradi[4], Gabriella Orlando[4], Antonella Santoro[4], Margherita Digaetano[4], Cinzia Puzzolante[4], Federica Carli[4], Andrea Bedini[4], Stefano Busani[4], Massimo Girardis[5], Andrea Cossarizza[3], Rossella Miglio[2,3], Cristina Mussini[1,4], Giovanni Guaraldi[1,4]*, Roberto D'Amico[2,3‡]

1 Department of Surgical, Medical, Dental and Morphological Sciences, University of Modena and Reggio Emilia, Modena, Italy, 2 Unit of Statistical and Methodological Support to Clinical Research, Azienda Ospedaliero-Universitaria, Modena, Italy, 3 Department of Medical and Surgical Sciences for Children and Adults, University of Modena and Reggio Emilia, Modena, Italy, 4 Infectious Disease Clinic, Azienda Ospedaliero-Universitaria di Modena, Modena, Italy, 5 Department of Anesthesia and Intensive Care, University of Modena and Reggio Emilia, Modena, Italy

‡ JM and FB share first authorship on this work. GG and RD are joint senior authors on this work.
* jovana.milic@gmail.com (JM); giovanni.guaraldi@unimore.it (GG)

## Abstract

### Background

The benefit of tocilizumab on mortality and time to recovery in people with severe COVID pneumonia may depend on appropriate timing. The objective was to estimate the impact of tocilizumab administration on switching respiratory support states, mortality and time to recovery.

### Methods

In an observational study, a continuous-time Markov multi-state model was used to describe the sequence of respiratory support states including: no respiratory support (NRS), oxygen therapy (OT), non-invasive ventilation (NIV) or invasive mechanical ventilation (IMV), OT in recovery, NRS in recovery.

### Results

Two hundred seventy-one consecutive adult patients were included in the analyses contributing to 695 transitions across states. The prevalence of patients in each respiratory support state was estimated with stack probability plots, comparing people treated with and without tocilizumab since the beginning of the OT state. A positive effect of tocilizumab on the probability of moving from the invasive and non-invasive mechanical NIV/IMV state to the OT in

**Data Availability Statement:** Data are available if requested at the EU platform of data sharing we apply to: https://www.ehden.eu.

**Funding:** The author(s) received no specific funding for this work.

**Competing interests:** The authors have declared that no competing interests exist.

recovery state (HR = 2.6, 95% CI = 1.2–5.2) was observed. Furthermore, a reduced risk of death was observed in patients in NIV/IMV (HR = 0.3, 95% CI = 0.1–0.7) or in OT (HR = 0.1, 95% CI = 0.0–0.8) treated with tocilizumab.

## Conclusion

To conclude, we were able to show the positive impact of tocilizumab used in different disease stages depicted by respiratory support states. The use of the multi-state Markov model allowed to harmonize the heterogeneous mortality and recovery endpoints and summarize results with stack probability plots. This approach could inform randomized clinical trials regarding tocilizumab, support disease management and hospital decision making.

## Background

COVID-19 pandemic has brought major challenges for health care systems even in well-organized and resource-rich settings. Although the evidence for effective treatments are still missing, the number of observational and randomized trials on antiviral drugs and immune-active agents is rapidly increasing [1].

Several randomized controlled trials assessing antiviral drugs in COVID-19 patients have recently used an outcome that was suggested by the WHO R&D Blueprint expert group [2–5]. This outcome is a time to event categorical variable that considers the occurrence of death, recovery, but also the switch of oxygen support states, which represent well the patients' pathway in hospitals and provide information on their prognosis [6].

This outcome depicts progression and recovery of COVID-19 patients by their need of different respiratory support aids. A patient is classified according the following categories: 1. not hospitalized with resumption of normal activities; 2. not hospitalized, but unable to resume normal activities; 3. hospitalized, not requiring supplemental oxygen; 4. hospitalized, requiring supplemental oxygen; 5. hospitalized, requiring nasal high-flow oxygen therapy, noninvasive mechanical ventilation, or both; 6. hospitalized, requiring ECMO, invasive mechanical ventilation, or both; and 7. death [6].

The WHO R&D Blueprint score has never been used to assess the benefit of tocilizumab in the treatment of COVID-19 pneumonia. Tocilizumab is an extensively used immune active agent which showed discordant results in the reduction of IMV and mortality in hospitalized patients with severe COVID-19 pneumonia [7–9]. TESEO cohort showed a significant reduction in risk of death (aHR = 0.38, 95% CI:0.17–0.83, p = 0.02) and/or mechanical ventilation (aHR = 0.61, 95% CI:0.40–0.92; p = 0.02) in patients who received tocilizumab over the standard of care [10]. On the contrary, initial randomized clinical trials, did not show any benefit [7, 11], while recent large randomized trials suggest reduced likelihood of progression to IMV and death in patients with severe COVID-19 pneumonia treated with tocilizumab [12–14]. In particular, an open-label randomized multicenter study conducted in Italy including 126 patients with $PaO_2/FiO_2$ between 200 and 300 mmHg did not show the benefit on disease progression [7]. On the other hand, REMAP-CAP trial showed improved 90-day survival in critically ill patients receiving organ support who were treated with tocilizumab or sarilumab (HR = 1.61, 95% CI: 1.25–2.08) [12]. Moreover, RECOVERY trial reports that patients receiving tocilizumab had improved survival and were less likely to reach the composite endpoint of IMV or death (33% vs. 38%; risk ratio 0.85; 95% CI 0.78–0.93; p = 0.0005) [14].

These conflicting results could be justified by diverse inclusion criteria, small sample size, competing effect of other treatments such as glucocorticoids, but also by different stages of the disease when tocilizumab is administered [15].

The latter could be of paramount importance. Therefore, we hypothesized that a multistate Markov model applied in accordance to WHO R&D Blueprint indication, may help to better depict the impact of tocilizumab at any disease stage as captured by respiratory aid change during hospitalization. We hereby present a secondary analysis of a subset of hospitalized patients treated with tocilizumab previously described in TESEO cohort.

The aim of this observational study was to estimate the impact of tocilizumab administration on switching respiratory support states, mortality and time to recovery in hospitalized patients with SARS-CoV-2 pneumonia.

## Methods

### Population

This study included consecutive adult patients ($\geq$18 years) admitted to Infectious Disease Clinic of the University Hospital of Modena, Italy, from 21 February to 10 April 2020, with radiological findings suggestive for SARS-CoV-2 pneumonia and confirmed by PCR method on nasopharyngeal swab, belonging to the previously described TESEO cohort [10], but restricting the analyses to the Modena's cohort subset only. Inclusion and exclusion criteria were the same of the parental cohort [10]. Patients that did not satisfy criteria for severe pneumonia were excluded from the analysis. Each patient was followed up until discharge or death or up to 10 April 2020.

### Definition of states characterising hospital clinical pathways

**Patients received oxygen supply to target SaO$_2$>90%.** Following the approach suggested by WHO R&D Blueprint expert group, we defined five possible consecutive respiratory support states, that are hereinafter reported and referred as intermediate states:

- State 1: no respiratory support (NRS);

- State 2: oxygen therapy (OT);

- State 3: non-invasive ventilation (NIV) or invasive mechanical ventilation (IMV);

- State 4: OT in recovery;

- State 5: NRS in recovery.

Two respiratory support states, such as non-invasive ventilation (NIV) or invasive mechanical ventilation (IMV) states were grouped to have a properly sized single category. Additionally, as per inclusion criteria only hospitalized patients were enrolled, we were not able to use the complete scale proposed by WHO R&D Blueprint expert group, which also comprises non-hospitalized patients.

Patients were admitted to the hospital at any state between 1 and 3 and during their hospital stay they did not necessarily switch through all states. The forward changes, which happen from any state between 1 to 3, indicated that patients were worsening, whereas those occurring from state 3 to 5 suggested that patients were improving.

Time was measured in minutes and reported in days, and was calculated as the elapsed time between the time the patient entered in one state and the time in which he/she moved forward to another one. Patients could leave the pathway for 2 possible reasons:

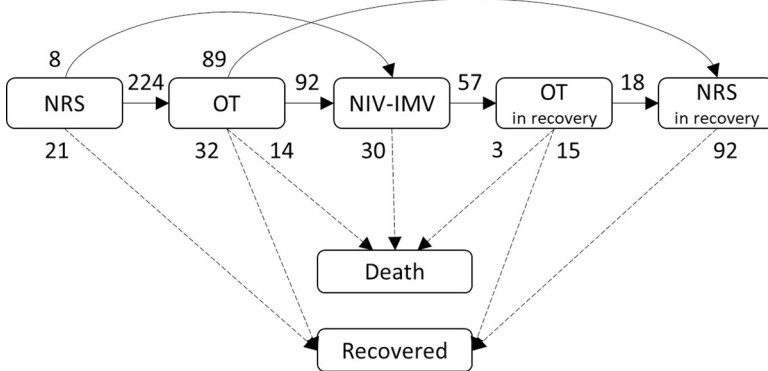

**Fig 1. Representation of clinical pathways and number of observed changes of state.** Notes: Arrows represent transitions between states and the number of observed transitions is reported. The dotted lines represent the final transitions of the study. NRS = No Respiratory Support; OT = Oxygen Therapy; NIV = Non-Invasive Ventilation; IMV = Invasive Mechanical Ventilation. 256 patients have started their hospitalization in the NRS state, 12 in the OT state and 3 in the NIV / INT state A total number of 64 patients with censored length of hospital stay were present: 3 in NRS, 9 in OT, 16 in NIV-IMV, 21 in OT in recovery, 15 in NRS in recovery.

1. discharge, which can be at home, or at post-acute care facility;

2. in-hospital death.

Censoring was defined as the condition in which the patient was either still hospitalized on 10 April 2020 at 12:00 AM or transferred to another acute care facility. The representation of this sequence of states is reported in Fig 1.

The main outcomes of the study were: progression from OT or NIV/IMV states to death, progression from OT to NIV/IMV, recovery from NIV/IMV state to the OT state. All the other changes in respiratory states were considered as secondary outcomes.

## Pharmacological interventions for COVID-19 pneumonia

All patients were treated with standard of care in agreement with the Regional Guidelines of Emilia Romagna [16] regarding the treatment of COVID-19 that were continuously updated during the period of the study. The treatments consisted of:

- Hydroxychloroquine (400 mg BID on day 1 followed by 200 mg BID on days 2 to 5 eventually adjusted for creatinine clearance estimated by a CKD algorithm);

- Azithromycin (500 mg QD for 5 days) at physician's discretion when suspecting a bacterial respiratory superinfection;

- Low molecular weight heparin for prophylaxis of deep vein thrombosis according to body weight and renal function.

- Lopinavir/ritonavir (400/100 mg BID) or darunavir/cobicistat (800/150 mg QD) for 14 days were used up to 18 March, when a clinical trial on the former did not show any benefit of protease inhibitors against the standard of care [3]. Other antiviral agents showing promising results such as remdesivir [17], was not used in our patients, due to its unavailability on the market.

Moreover, some patients were treated with immune-active agents, such as tocilizumab and glucocorticoids, that were not standard of care during observational period. Tocilizumab was offered by a treating physician to patients who experienced arterial oxygen saturation

SaO$_2$<93% and a PaO$_2$/FiO$_2$<300 mmHg in room air or a decrease in PaO$_2$/FiO$_2$ > 30% in the previous 24 hours during hospitalization. All tocilizumab-treated patients provided a consent to participate in the study. Tocilizumab was administered intravenously (IV) or subcutaneously (SC) depending on hospital availability of the formulation at time of treatment [10]. IV tocilizumab was administered at the dose of 8 mg/kg of body weight (up to a maximum dosage of 800 mg) repeated twice, 12 hours apart, while SC formulation at the dose of 162 mg administered twice simultaneously, one per each thigh [18]. The patients who refused tocilizumab were treated with standard of care only.

Glucocorticoids were administered alone when tocilizumab was not available. On the contrary, patients underwent glucocorticoids after tocilizumab in the case of treatment failure, defined as lack of improvement of PaO$_2$/FiO$_2$ after three days of tocilizumab administration, and these patients were included in the tocilizumab group. Patients who received anakinra or other immunomodulatory agents other than tocilizumab or glucocorticoids were excluded from the analysis.

All patients were evaluated possible pharmacologic drug-drug interactions before administration of any treatment and in particular cases, dosages were adjusted according to other co-morbidities and co-medications. Patients were also carefully monitored for potential toxicities of each pharmacological agent.

## Covariates

All the data was obtained from routinely collected medical records. The patients' full medical history, demographic and epidemiological data as well as the value of PaO$_2$/FiO$_2$ at baseline were obtained at the hospital admission. The risk of multiorgan failure and mortality was assessed with standardized Subsequent Organ Failure Assessment (SOFA) score. Other covariates including age, gender, days from epidemic onset (defined as 21 February as the date of first COVID-19 case in the Modena province) were considered. Clinical data with patients' signs, symptoms, blood count, coagulation, inflammatory and biochemical markers were routinely collected and reported in an electronic patient charts. Other pharmacological interventions were glucocorticoids and lopinavir/ritonavir or darunavir/cobicistat. Use of glucocorticoids was considered as a potential confounder in adjusted models.

## Statistical analysis

Continuous variables were described as mean or median and inter-quartile range (IQR), whereas categorical variables summarized as absolute and percentage frequencies. Comparison among patients receiving different types of immune-active agents were assessed by using the Yates chi-square test for categorical variables or the Wilcoxon test for quantitative variables.

Statistical analyses were carried out by using a continuous-time Markov multi-state model, which considered the previously defined sequence of respiratory support states. The transition probabilities, that is the probability to switch from a state to another one, were modelled according to an exponential distribution for time-to-event data, which takes into account censored follow-up times. Time was measured in days, considering the elapsed time in each transition across states. The Markov multi-state approach, compared to classical statistical methods for time-to-event data, has the advantage of allowing the joint analysis of length of care and incidence of clinical outcomes, such as the need for respiratory support or mortality, providing a complete assessment of COVID-19 inpatients' progress [19]. Under a statistical perspective this approach is also preferable as it can accommodate for competing outcomes, such as death or discharge, as well as for time-dependent multiple events, such as the need for respiratory support states [20].

At the beginning an unadjusted multi-state model analysis was carried out to assess the impact of tocilizumab on changes of state. Stratified predicted probabilities for all changes of states were then estimated, as well as the predicted lengths of stay in each state. These predictions were calculated for patients receiving tocilizumab since the beginning of the OT state and for patients never receiving it. A multivariable multi-state model was then considered, where the associations between the administration of tocilizumab and the changes of states were adjusted for the following variables: age, gender, SOFA score at hospital admission, time from the onset of COVID-19 in Modena and administrations of glucocorticoids and lopinavir/ritonavir or darunavir/cobicistat. Unadjusted and adjusted hazard ratios (HR) were used to measure the associations between tocilizumab administration and changes of state.

Treatment with azithromycin, hydroxychloroquine and low molecular weight heparin were not included in the multivariable model, as they were administered to all the patients. Time from COVID-19 onset in Modena was measured as the difference in days between the date of hospital admission and the 21 February 2020. The effect of age was reported as 5-years linear increment, the impact of SOFA score was described as 1-point linear increment, whereas the one related to time from COVID-19 onset as 1-week linear increment. The associations between the independent variables and the change of respiratory support states were assessed only for those transitions observed in at least 10 patients, whereas for the remaining transitions no independent variables were assessed in the models.

To properly assess the effect of tocilizumab on outcomes, its administration was modelled as a time-dependent covariate. This allowed to assess the effect starting from the exact time of first administration in each patient and to prevent the occurrence of immortal time bias. The effect of drugs administration was only estimated from the OT state onwards. Age, gender, SOFA score, time from the onset of COVID-19 in Modena and other pharmacological interventions were also evaluated as potential effect modifiers by adding interaction terms in the multivariable multi-state model.

To visualize the results simultaneously over time in a single informative figure, we used stacked probability plots [19]. These were calculated by using the unadjusted multi-state model's predicted prevalence of patients over time in each state, including discharge and mortality. The plots were referred to: patients who were treated with tocilizumab from the OT state onwards; patients who were not treated with tocilizumab during hospitalization.

HRs and predicted probabilities of change of respiratory support states were reported as point estimates and as 95% confidence intervals (95% CI). The number needed to treat was also calculated for those changes of state that describe a clinical worsening and for which a statistically significant results was observed. The goodness of fit of the models was assessed by comparing the observed and expected prevalence across time, for all respiratory support states, including hospital discharge and death. The analyses were performed by using R 3.6.3 statistical software (The R Foundation for Statistical Computing, Wien). The significance level was set to 0.05, except for HRs, which were tested at both $p < 0.05$ and $p < 0.01$ levels.

## IRB approval and data information

Patients who received tocilizumab provided verbal, not written, informed consent because of isolation precautions. This procedure was approved by the Regional Ethical Committee of Emilia Romagna (protocol number: AOU 0018046/20). The same body also approved the retrospective analysis of the data and all patients provided informed consent to participate in the study. Oral consent was documented in the electronic patients' charts. Use of tocilizumab was based on clinical judgment and criteria reported in the TESEO study [10]. All patients with severe pneumonia admitted to hospital were considered for the analysis. Patients who

consented to use of tocilizumab, but rejected to participate in the study were excluded from the analysis. Patients were admitted from 21 February 2020 to 10 April 2020. The retrospective data were fully anonymized and the only sensitive data was year of birth.

## Results

### Transitions between states

Two hundred seventy-one patients were included in the analyses. The majority of patients were males (75.6%), the median age was 63 years. The overall follow-up time was 3159.8 total person-days (average 11.7, range 0.2–33.8) and 695 transitions across states were observed. On 10 April 2020 at 12:00 AM, 47 (17.3%) patients died, 160 (59.0%) recovered and 64 (23.6%) were still hospitalized or were transferred to another acute care facility. During the observed hospitalization, 256 (94.5%) patients were in the NRS state, 236 (87.1%) patients underwent OT, 103 (38.0%) NIV/IMV, 57 (21.0%) OT in recovery and 107 (39.5%) NRS in recovery. The number of states transitions is described in Fig 1. Respiratory impairment, expressed by $PaO_2/FiO_2$ had a median value of 229 (IQR 120–288) mmHg.

### Unadjusted effect of tocilizumab

During hospitalization, 122 (45.0%) patients received tocilizumab. The median time from admission to administration was 1.8 days (IQR 0.8–3.0). Tocilizumab was administered to 18 (14.8%) patients during NRS state, 78 (63.9%) during OT, 22 (18.0%) during NIV/IMV, 3 (2.5%) during OT in recovery and 1 (0.8%) during NRS in recovery.

No difference was found in age and gender between groups. The demographic and clinical characteristics of patients are reported in Table 1. Based on the observed data, some differences could be noticed in the two treatment groups.

Lower $PaO_2/FiO_2$ was observed in people treated with tocilizumab (164 mmHg, IQR 98–249) compared to those who were not treated (279, IQR 181–332). Patients treated with tocilizumab also had higher baseline SOFA score (3, IQR 2–4 vs 1, IQR 0–4)) and more altered markers of disease severity, namely lower lymphocytes (1000 c/mm$^3$, IQR 700–2200 vs 1400, IQR 900–2900) and higher CRP (12.3 mg/dL, IQR 5.6–17.7 vs 6.7, IQR 3.0–16.1). Furthermore, administration of glucocorticoids was more frequent among those who received tocilizumab (45.1% vs 15.4%), whereas darunavir/cobicistat was more frequent among those who did not receive tocilizumab (32.8% vs 52.3%). In this cohort, 3 patients only received lopinavir/ritonavir. The median times from hospitalization to administration of glucocorticoids and darunavir/cobicistat were 4.5 (IQR 1.5–8.8) and 0.3 (IQR 0.1–1.1) days, respectively. Glucocorticoids were mainly given during the OT (34.6%), NIV/IMV (26.9%) or OT in recovery (26.9%) states, whereas darunavir/cobicistat was often administered in earlier stages (47.5% in NRS and 44.9% in OT states).

Amongst patients who received tocilizumab 9 (7.4%) have died, whereas 38 (25.5%) patients have died without receiving tocilizumab. Considering patients who were given tocilizumab during NRS or OT states, 31 (32.3%) reached the NIV/IMV state. Conversely, amongst those who started their hospitalization in the NRS or OT states and were not given tocilizumab in these states, 69 (40.1%) underwent NIV or IMV.

As reported in Table 2, the administration of tocilizumab had a significant effect on the risks of change of state. Of note, administration of tocilizumab reduced the probability of transition from the OT state to death (1.2% vs 9.2%, HR = 0.1, 95% CI = 0.0–0.8) and from the NIV / IMV state to death (15.0% vs 55.6%, HR = 0.4, 95% CI = 0.2–0.9). Based on these data, the number needed to treat was equal to 12.5 and 2.5, respectively. Moreover, tocilizumab was associated with an increased probability of switching from the NIV / IMT state to the OT in

**Table 1. Descriptive characteristics of patients at hospital admission.**

| | Available data (n) | Tocilizumab during hospitalization (n = 122) | No Tocilizumab (n = 149) | p[1] |
|---|---|---|---|---|
| Age, years, median (IQR) | 271 | 63 (54–72) | 63 (53–74) | 0.813 |
| Male sex, % | 271 | 91 (74.6%) | 114 (76.5%) | 0.823 |
| Days from symptoms onset, median (IQR) | 124 | 7.0 (3.5–9.1) | 6.2 (3.3–8.1) | 0.661 |
| Weeks from COVID19 onset in Italy, median (IQR) | 271 | 4.1 (3.3–4.8) | 2.9 (2.1–4.1) | 0.000 |
| BMI, kg/m$^2$, median (IQR) | 114 | 28.1 (25.3–31.2) | 26.7 (24.4–29.7) | 0.133 |
| SOFA score, median (IQR) | 260 | 3 (2–4) | 1 (0–4) | 0.000 |
| Respiratory rate, median (IQR) | 199 | 25 (20–30) | 20 (17–24) | 0.000 |
| pH, median (IQR) | 222 | 7.5 (7.4–7.5) | 7.5 (7.4–7.5) | 0.223 |
| PaCO$_2$, mmHg, median (IQR) | 222 | 35.9 (32.6–39.9) | 36.5 (33.2–39.9) | 0.531 |
| PaO$_2$, mmHg, median (IQR) | 222 | 62.0 (55.7–71.2) | 65.0 (56.5–77.2) | 0.158 |
| SaO$_2$, mmHg, median (IQR) | 221 | 93.3 (90.9–95.2) | 94.0 (90–95.9) | 0.688 |
| PaO$_2$/FiO$_2$, mmHg, median (IQR) | 207 | 164 (98–249) | 279 (181–332) | 0.000 |
| White blood cells, c/mm$^3$, median (IQR) | 244 | 6185 (5150–8338) | 6040 (4950–8575) | 0.902 |
| Hemoglobin, g/dL, median (IQR) | 245 | 13.3 (12.2–14.1) | 13.3 (11.4–14.4) | 0.967 |
| Lymphocytes, c/mm$^3$, median (IQR) | 145 | 1000 (700–2200) | 1400 (900–2900) | 0.048 |
| Platelets, x 10$^9$, median (IQR) | 244 | 215 (166–266) | 202 (148–271) | 0.623 |
| Bilirubin, mg/dL, median (IQR) | 233 | 0.6 (0.5–0.8) | 0.6 (0.5–0.8) | 0.375 |
| Creatinine, mg/dL, median (IQR) | 244 | 0.8 (0.7–1.1) | 0.9 (0.7–1.1) | 0.697 |
| Aspartate aminotransferase, IU/L, median (IQR) | 244 | 35 (22–59) | 30 (20–44) | 0.043 |
| Creatine kinase, IU/L, median (IQR) | 242 | 126 (60–260) | 101 (50–256) | 0.335 |
| D-dimer, mg/mL, median (IQR) | 242 | 1150 (745–2133) | 1405 (625–2565) | 0.628 |
| Lactate dehydrogenase, IU/L, median (IQR) | 243 | 628 (533–782) | 531 (419–790) | 0.001 |
| C-reactive protein, mg/dL, median (IQR) | 244 | 12.3 (5.6–17.7) | 6.7 (3.0–16.1) | 0.002 |
| Interleukin-6, pg/mL, median (IQR) | 167 | 202 (109–443) | 194 (57–366) | 0.275 |
| Procalcitonin, ng/mL, median (IQR) | 168 | 0.1 (0.1–0.3) | 0.2 (0.1–2) | 0.007 |
| Troponin, ng/L, median (IQR) | 105 | 12 (12–21) | 12 (12–27) | 0.876 |
| Fibrinogen, mg/dL, median (IQR) | 141 | 647 (509–733) | 606 (505–709) | 0.419 |
| Use of glucocorticoids, % | 271 | 55 (45.1%) | 23 (15.4%) | 0.000 |
| Use of darunavir/cobicistat, % | 271 | 40 (32.8%) | 78 (52.3%) | 0.002 |

Notes: median (inter-quartile range) were reported for quantitative variables and n (%) were reported for categorical variables.

[1] = Wilcoxon rank sum test for quantitative variables and Yates chi-square test for categorical variables.

recovery state (85.0% vs 44.4%, HR = 2.7, 95% CI = 1.6–4.7). The predicted lengths of stay in each state, for patients receiving and not receiving tocilizumab, were as follows: 6.2 (95% CI = 5.2–7.9) and 5.7 (95% CI = 4.7–6.6) days in OT; 7.3 (95% CI = 5.5–9.7) and 10.4 (95% CI = 7.6–14.1) days in NIV/IMV; 12.7 (95% CI = 8.7–18.7) and 11.3 (95% CI = 6.3–20.2) days in OT in recovery.

The prevalence of patients in each respiratory support state was estimated in the stack probability plot, comparing people treated with and without tocilizumab since the beginning of the OT state (Fig 2).

### Adjusted effect of tocilizumab

The results of the multivariable multi-state model, adjusted for age, gender, SOFA score, time from the onset of COVID-19 epidemic and administrations of glucocorticoids and lopinavir/ritonavir or darunavir/cobicistat, were reported in Table 3. This analysis was carried out considering 260 patients with complete data.

**Table 2. Unadjusted effect of tocilizumab use on change of state risks.**

| From | To | Tocilizumab Yes | | Tocilizumab No | | Tocilizumab Yes vs No | |
|---|---|---|---|---|---|---|---|
| | | Predicted probability | 95% CI | Predicted probability | 95% CI | HR | 95% CI |
| NRS | OT | 88.6% | 83.9–91.7% | 88.6% | 83.9–91.7% | - | - |
| | NIV / IMV | 3.2% | 1.5–6.3% | 3.2% | 1.5–6.3% | - | - |
| | Recovered | 8.3% | 5.6–12.4% | 8.3% | 5.6–12.4% | - | - |
| OT | NIV / IMV | 33.8% | 23.9–44.0% | 44.7% | 36.3–52.6% | 0.66 | 0.42–1.02 |
| | NRS in recovery | 44.3% | 33.8–54.6% | 36.2% | 28.3–43.9% | 1.07 | 0.70–1.62 |
| | Recovered | 20.7% | 13.4–30.6% | 9.9% | 6.2–16.1% | 1.82 | 0.91–3.67 |
| | Death | 1.2% | 0.2–8.8% | 9.2% | 5.3–15.3% | 0.11 # | 0.01–0.83 |
| NIV/IMV | OT in recovery | 85.0% | 72.7–92.4% | 44.4% | 30.7–60.0% | 2.72 ## | 1.56–4.75 |
| | Death | 15.0% | 7.6–27.3% | 55.6% | 40.0–69.3% | 0.38 # | 0.16–0.89 |
| OT in recovery | NRS in recovery | 50.8% | 31.5–67.8% | 51.8% | 22.7–76.7% | 0.93 | 0.33–2.61 |
| | Recovered | 40.9% | 24.0–59.5% | 40.3% | 16.9–69.6% | 0.96 | 0.30–3.02 |
| | Death | 8.3% | 2.6–23.7% | 7.9% | 2.1–23.7% | - | - |
| NRS in recovery | Recovered | 100.0% | - | 100.0% | | 0.98 | 0.65–1.48 |

Notes: NRS = No Respiratory Support; OT = Oxygen Therapy; NIV = Non-Invasive Ventilation; INT = Intubation; HR = Hazard Ratio; 95% CI = 95% Confidence Interval. Predicted probabilities and HRs were estimated by using a Markov multi-state model. The predicted probabilities were referred to: patients who were treated with tocilizumab from the OT state onwards; patients who were not treated with tocilizumab during hospitalization. This analysis was not adjusted for confounders.
# = statistically significant at 95% confidence level (p < 0.05)
## = statistically significant at 99% confidence level (p < 0.01).

As in the unadjusted analysis, there was a positive effect of tocilizumab on the probability of moving from the NIV / IMV state to the OT in recovery state (HR = 2.6, 95% CI = 1.2–5.2). Furthermore, a reduced risk of death was observed in patients in NIV / IMV state (HR = 0.3, 95% CI = 0.1–0.7) or in OT (HR = 0.1, 95% CI = 0.0–0.8) treated with tocilizumab.

Time from the onset of COVID-19 epidemic was not associated with any risk of change of respiratory support state, whereas higher age, male gender and higher SOFA score were

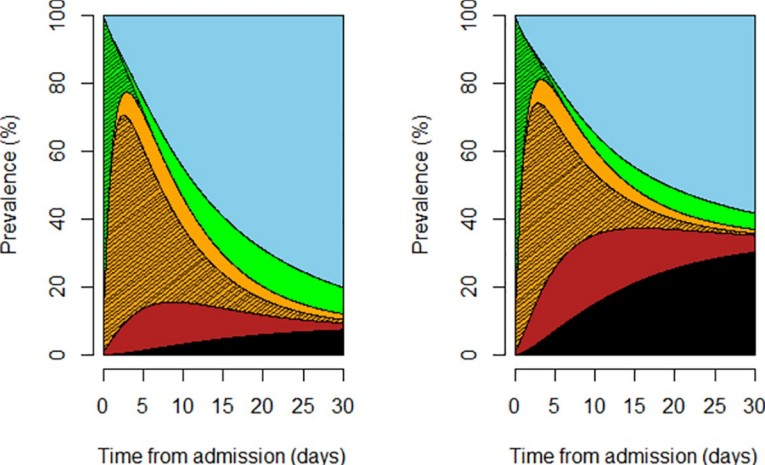

**Fig 2. The predicted prevalence of patients in each respiratory support state was estimated in the stack probability plot, comparing patients who were treated with tocilizumab from the OT state onwards (left plot) and patients who were not treated with tocilizumab during hospitalization (right plot).** Color code: black = death; blue = discharge; red = non-invasive ventilation/invasive mechanical ventilation; orange = oxygen therapy; green = no respiratory support. The strikethrough area refers to NRS or OT, whereas the "clean" areas refer to NRS or OT in recovery.

**Table 3. Adjusted effect of administration of tocilizumab on change of state risks.**

| Change of state [1] | Tocilizumab Yes vs No | |
|---|---|---|
| | HR | 95% CI |
| From OT to NIV / IMV | 0.67 | 0.42–1.08 |
| From OT to NRS in recovery | 1.40 | 0.86–2.26 |
| From OT to Recovered | 1.96 | 0.91–4.24 |
| From OT to Death | 0.12 # | 0.01–0.93 |
| From NIV / IMV to OT in recovery | 2.66 ## | 1.27–5.57 |
| From NIV / IMV to Death | 0.30 # | 0.12–0.77 |
| From OT in recovery to NRS in recovery | 0.90 | 0.26–3.19 |
| From OT in recovery to Recovered | 1.79 | 0.40–7.95 |
| From NRS in recovery to Recovered | 1.36 | 0.86–2.17 |

Notes: NRS = No Respiratory Support; OT = Oxygen Therapy; NIV = Non-Invasive Ventilation; IMV = Invasive Mechanical Ventilation; HR = Adjusted Hazard Ratio; 95% CI = 95% Confidence Interval; The multivariable Markov multi-state model considered age, gender, SOFA score, time from COVID onset in Modena, administration of tocilizumab, glucocorticoids and lopinavir/ritonavir or darunavir/cobicistat as independent variables. [1] = the effect of independent variables was assessed only for changes of state that were observed in at least 10 patients
# = statistically significant at 95% confidence level ($p < 0.05$)
## = statistically significant at 99% confidence level ($p < 0.01$)

associated with several changes of respiratory state, all of them in the direction of a worse prognostic pathway (S1 Table).

We also observed an increased mortality in patients treated with glucocorticoids when in the OT state and in patients treated with lopinavir/ritonavir or darunavir/cobicistat when in the NIV / IMV state (S1 Table). There was no evidence of modification of the effect of tocilizumab, due to age, gender, SOFA score, time from Covid-19 onset and other pharmacological interventions, based on significance testing of interaction terms.

## Discussion

This prospective observational study showed the beneficial effect of tocilizumab in reducing the risk of death in patients with COVID-19 pneumonia. The reduced risk of death was larger at the initial oxygen support state, although this finding was based on a smaller number of patients, and persisted throughout advanced NIV/IMV support, respectively of 88% when used in OT and 70% in the NIV/IMV. Furthermore, patients receiving tocilizumab had an increased probability of recovery when in NIV/IMV state.

The reduced risk of death was previously described by our group in the context of the larger TESEO cohort [10] and we confirm, with a more detailed statistical approach, the lower mortality rate.

A systematic review and meta-analysis of 10 observational studies including 1358 patients demonstrated that mortality was 12% lower for COVID-19 patients treated with tocilizumab compared to patients who were not treated with tocilizumab. The number needed to treat was 11, suggesting that for every 11 COVID-19 patients treated with tocilizumab 1 death was prevented.

Moreover, encouraging results from large randomized clinical trials [12, 14] could provide a greater level of evidence and recommendations in the current guidelines for the treatment of COVID-19 pneumonia.

This multi-state model allowed to depict a different beneficial effect size of tocilizumab according to different respiratory support states. We observed that tocilizumab may have had

a higher impact when used in the early disease stage, not only during IMV and pronounced cytokine storm. Tocilizumab blocks IL-6 receptors and along with TNF-alpha and IL-1 is involved in the pathways of cytokines' cascade [21]. In the late phase of acute respiratory distress syndrome (ARDS), when also other immunological factors are involved in pathogenesis of severe COVID-19 pneumonia, this agent might not be sufficient to face an uncontrolled inflammatory response [22].

The multivariable model did not include hydroxychloroquine, azithromycin and low molecular weight heparin because used as standard of care in all patients. In consideration of the fact that lopinavir/ritonavir or darunavir/cobicistat were administered to all the patients only before 18 March 2020, it was also decided to correct the multivariate analyses according to date of onset of the epidemic. Given the design of the study, we could not assess the role of tocilizumab without standard of care, therefore, we argue that tocilizumab alone may not be sufficient as a treatment of severe COVID-19 pneumonia. It is out of the purpose of this study to describe adverse events potentially associated with tocilizumab use, as being previously described in detail in the TESEO cohort [10].

The major confounders analyses in this study played a significant role in the progression of respiratory support states and therefore prevent from the most relevant sources of confounding bias. As previously described, age was a major driver of mortality in any disease stage [23]. Interestingly, age also had a negative impact in disease recovery. This is clinically evident in the loss of functional ability related to the intense catabolic damage and lean mass loss with progression of frailty [24]. Various studies reported a higher mortality rate among males [23, 25]. However, the underlying mechanism is still unknown. It has been suggested that steroids and activity of X-linked genes can modulate the immune response to viral infection [26, 27]. Additionally, ACE2 receptors might have a higher expression in men [28].

SOFA was included in the model as an indicator of hypertension, cardiovascular disease, chronic renal insufficiency or cancer. These disease conditions are potentially associated with organ failure during the acute inflammatory states induced by COVID-19 and, as expected they interfered with both intervention and outcome risks.

As previously described by our group, patients' trajectories across subsequent respiratory support states well depict the natural history and prognosis in hospitalized patients with COVID-19 pneumonia [submitted paper]. We propose a multi-state model that predicts a daily evolution of respiratory-support aids in patients with COVID-19 pneumonia and can be used to identify patients who will likely benefit most from immuno-active drugs. The application of this Markov model has been widely used in the setting of chronic disease conditions and have been rarely applied to the rapidly evolving clinical evolution of COVID-19 pneumonia [19].

Our statistical approach accommodates the primary analysis with a stacked probability plot of the major events such as being discharged alive and death. This model is a powerful tool to describe clinically opposite endpoints (mortality, discharge) as well as endpoints influencing hospital capacities (duration of hospitalization) simultaneously over time. In particular, the stack probability plots visualize results over time in a single informative plot. These figures allow the joint evaluation of multiple endpoints by visual comparison of depicted areas.

As described by Von Cube et al [6], the heterogeneity in choice of endpoints reflects the manifold potential treatment effects, the various different decision makers involved and the different patient populations under study. Nonetheless, harmonization of the different endpoints is an essential step to allow comparison of results from clinical trials and fasten decision making.

We acknowledge some limitations of our study. Firstly, we cannot exclude residual confounding bias by indication due to patients' characteristics that we did not adjust for, such as

body mass index or comorbidities. However, by including clinically meaningful covariates such as age, gender and risk of multi-organ failure (SOFA) in the multivariable model, these bias can only be caused by other characteristics and should be of acceptable magnitude. More-over, we also controlled for confounding related to time from COVID-19 onset, since in this study there were patients treated at the very beginning of the outbreak as well as several weeks later. Secondly, results may be partly influenced by the assumptions of the multi-state model, namely those concerning the onset and the duration of clinical effect of tocilizumab. Thirdly, this study was performed in one single tertiary centre in Italy and may thus not reflect general management of SARS CoV-2 pneumonia in other settings. Fourthly, NIV and IMV were grouped together in order to have properly sized single category, but this choice did not permit to distinguish the escalation from NIV to IMV and de-escalation from IMV to NIV as impor-tant steps in management of severe COVID-19 pneumonia. Fifthly, multistate models have the limitation that they do not account for patient triage possibly due to ICU congestion. Addi-tional research is needed to avoid selection bias arising from this special clinical situation. This might be due to different epidemic environments over countries and due to heterogeneity in hospital health care capacity.

To conclude, we were able to show the positive impact of tocilizumab used in different dis-ease stages depicted by respiratory support states. The use of the multi-state Markov model allowed to harmonize the heterogeneous mortality and recovery endpoints and summarize results with stack probability plots. This approach could inform randomized clinical trials regarding tocilizumab, support disease management and hospital decision making.

## Supporting information

**S1 Table. Adjusted effect of other covariates on change of state risks.**
(DOCX)

## Acknowledgments

The authors would like to thank Fogliani Rossella, Righini Grazia and Lugli Mario for their contribution in the data collection. Without them this paper would not be possible.

## Author Contributions

**Conceptualization:** Jovana Milic, Federico Banchelli, Marianna Meschiari, Giacomo Ciusa, Rossella Miglio, Cristina Mussini, Giovanni Guaraldi, Roberto D'Amico.

**Data curation:** Jovana Milic, Federico Banchelli, Marianna Meschiari, Erica Franceschini, Giacomo Ciusa, Licia Gozzi, Sara Volpi, Matteo Faltoni, Giacomo Franceschi, Vittorio Iadi-sernia, Dina Yaacoub, Giovanni Dolci, Carlotta Rogati, Marco Tutone, Giulia Burastero, Alessandro Raimondi, Marianna Menozzi, Gianluca Cuomo, Luca Corradi, Gabriella Orlando, Antonella Santoro, Margherita Digaetano, Cinzia Puzzolante, Federica Carli, Andrea Bedini, Stefano Busani, Massimo Girardis, Andrea Cossarizza, Giovanni Guaraldi.

**Formal analysis:** Jovana Milic, Federico Banchelli, Erica Bacca, Rossella Miglio, Roberto D'Amico.

**Investigation:** Jovana Milic, Federico Banchelli, Marianna Meschiari, Erica Franceschini, Gia-como Ciusa, Licia Gozzi, Sara Volpi, Matteo Faltoni, Giacomo Franceschi, Vittorio Iadiser-nia, Dina Yaacoub, Giovanni Dolci, Erica Bacca, Carlotta Rogati, Marco Tutone, Giulia Burastero, Alessandro Raimondi, Marianna Menozzi, Gianluca Cuomo, Luca Corradi, Gabriella Orlando, Antonella Santoro, Margherita Digaetano, Cinzia Puzzolante, Federica

Carli, Andrea Bedini, Stefano Busani, Massimo Girardis, Andrea Cossarizza, Rossella Miglio, Cristina Mussini, Giovanni Guaraldi.

**Methodology:** Jovana Milic, Federico Banchelli, Rossella Miglio, Cristina Mussini, Giovanni Guaraldi, Roberto D'Amico.

**Resources:** Giovanni Guaraldi.

**Supervision:** Jovana Milic, Federico Banchelli, Cristina Mussini, Giovanni Guaraldi, Roberto D'Amico.

**Validation:** Jovana Milic, Federico Banchelli, Cristina Mussini, Giovanni Guaraldi, Roberto D'Amico.

**Visualization:** Jovana Milic, Federico Banchelli, Cristina Mussini, Giovanni Guaraldi, Roberto D'Amico.

**Writing – original draft:** Jovana Milic, Federico Banchelli, Marianna Meschiari, Erica Franceschini, Giovanni Guaraldi, Roberto D'Amico.

**Writing – review & editing:** Jovana Milic, Federico Banchelli, Cristina Mussini, Giovanni Guaraldi, Roberto D'Amico.

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
