## [Decision Letter · Decision Letter 0]

2 Feb 2021

PONE-D-20-37536

The impact of tocilizumab on respiratory support states transition and clinical outcomes in COVID-19 patients. A Markov model multi-state study

PLOS ONE

Dear Dr. Milic,

Thank you for submitting your manuscript to PLOS ONE. After careful consideration, we feel that it has merit but does not fully meet PLOS ONE’s publication criteria as it currently stands. Therefore, we invite you to submit a revised version of the manuscript that addresses the points raised during the review process.

We look forward to receiving your revised manuscript.

Kind regards,

Chiara Lazzeri

Academic Editor

PLOS ONE

Journal Requirements:

2. In the Ethics Statement on the online submission form and the manuscript Methods , please clarify the context in which consent was obtained, and specify whether patients provided:

    1) Consent to use their medical records/samples used in research

    2) Consent to receive tocilizumab

    3) Consent to take part in the study reported in this manuscript.

If your study involved retrospective collection of data from patient medical records, please ensure that you have discussed whether all data/samples were fully anonymized before you accessed them and/or whether the IRB or ethics committee waived the requirement for informed consent. If patients provided informed written consent to have data/samples from their medical records used in research, please include this information.

Reviewers' comments:

Reviewer's Responses to Questions

**Comments to the Author**

1. Is the manuscript technically sound, and do the data support the conclusions?

Reviewer #1: Partly

Reviewer #2: Yes

2. Has the statistical analysis been performed appropriately and rigorously? 

Reviewer #1: I Don't Know

Reviewer #2: Yes

3. Have the authors made all data underlying the findings in their manuscript fully available?

Reviewer #1: Yes

Reviewer #2: Yes

4. Is the manuscript presented in an intelligible fashion and written in standard English?

Reviewer #1: Yes

Reviewer #2: Yes

5. Review Comments to the Author

Reviewer #1: The authors submitted the report of an interesting nonrandomized study on the role of tocilizumab in patients with covid-19.

The manuscript is well written, and tocilizumab is of recent interest as experimental treatment for covid-19, with conflicting and unclear evidence. New data would be important, but I have some issues that can be better addressed by the authors.

I hope my comments would be of help.

Below, my specific comments.

ABSTRACT

1) I would move information on study design to the methods subsection of the abstract, providing here background brief information and aim.

BACKGROUND

2) Background should be widened, providing the reader a summary of the conflicting evidence available on tocilizumab. The authors may cite updated reviews and/or the results of recent literature on the topic (e.g. doi: 10.1016/j.pulmoe.2020.07.003; doi: 10.1056/NEJMoa2030340 ; doi: 10.1016/S2665-9913(20)30313-1; COVACTA trial and EMBACTA trial).

METHODS

1) Please specify if the patients’ cohort here described overlaps with the one of the TESEO study. If so, the authors should clarify this.

2) Please specify whether every admitted patient in the study period was enrolled and the eventual exclusion criteria.

3) Why did the authors not decide to use the complete WHO clinical progression scale as outcome?

4) “Two respiratory support states, such as non-invasive ventilation (NIV) or invasive mechanical ventilation (IMV) states were grouped to have a properly sized single category”. This is an important limitation of the study, as it is impossible to distinguish the escalation from NIV to IMV but also the de-escalation from IMV to NIV. The authors should address the limitation and, if possible, provide alternative analysis with the two distinct categories.

5) In the covariates subparagraph, the authors should be express more clearly if glucocorticoids were considered potential confounders and the analysis was controlled for their administration.

DISCUSSION

1) The authors may consider providing brief mentioning current guidelines recommendation towards or against tocilizumab for covid19

CONCLUSIONS

2) The authors conclude saying that “we were able to show the positive impact of tocilizumab used in different disease stages, especially in the early onset of ARDS”. I would change the term “ARDS” with “COVID-19”. Furthermore, I would rephrase the conclusions with a more conservative approach. Data are still conflicting and the nonrandomized study design suggest not to over rely on such results, while awaiting for further RCTs.

Reviewer #2: In the paper by J. Milic, the authors analyze the results of tocilizumab therapy on a cohort of covid-19 affected patients, already described on a previous prospective observational study (Guaraldi G eta al 2020). In this second paper they use a different statistical approach: the Markov multy-state model. The tocilizumab group, compared with the control group, clearly show worst clinical conditions at the time of enrollment, with worst respiratory function, higher flogosis parameters and lower lymphocytes counts, nevertheless the tocilizumab group presents a better survival and in general a better recovery. For these reasons, in the opinion of the reviewer, the starting differences between the two groups do not impair the results.

As the authors asses in the discussion section, the scientific production on this field is in part controversial, in particular they refer to the paper by Salvarani C et al 2020. Have the authors any possible explanation or comments on these contrasting results?

In the result section the sentence about the lymphocytes counts in the two group seems to be inverted: 1.0 x 106/L should refer to tocilizumab group, as indicated in the table not to the control group as stated in the text.

6. PLOS authors have the option to publish the peer review history of their article (what does this mean?). If published, this will include your full peer review and any attached files.

Reviewer #1: No

Reviewer #2: No

---

## [Author Response · Author response to Decision Letter 0]

23 Apr 2021

Modena, 27 February 2021

Dear Editor, 

We are very grateful for your constructive comments and suggestions to our paper entitled: “The impact of tocilizumab on respiratory support states transition and clinical outcomes in COVID-19 patients. A Markov model multi-state study”.

We here provide a point-by-point reply to the comments and we have incorporated the related changes in the manuscript. We thank the reviewers for their thoughtful insights which helped to significantly improve the manuscript.

Reviewer #1: The authors submitted the report of an interesting nonrandomized study on the role of tocilizumab in patients with covid-19.

The manuscript is well written, and tocilizumab is of recent interest as experimental treatment for covid-19, with conflicting and unclear evidence. New data would be important, but I have some issues that can be better addressed by the authors.

I hope my comments would be of help.

Below, my specific comments.

ABSTRACT

1) I would move information on study design to the methods subsection of the abstract, providing here background brief information and aim.

Authors’ answer:

The following sentence has been changed:

“The benefit of tocilizumab on mortality and time to recovery in people with severe COVID pneumonia may depend on appropriate timing. The objective was to estimate the impact of tocilizumab administration on switching respiratory support states, mortality and time to recovery.”

BACKGROUND

2) Background should be widened, providing the reader a summary of the conflicting evidence available on tocilizumab. The authors may cite updated reviews and/or the results of recent literature on the topic (e.g. doi: 10.1016/j.pulmoe.2020.07.003; doi: 10.1056/NEJMoa2030340 ; doi: 10.1016/S2665-9913(20)30313-1; COVACTA trial and EMBACTA trial).

Authors’ answer:

As suggested by the reviewer, the background regarding randomized clinical trial on tocilizumab has been updated:

“TESEO cohort showed a significant reduction in risk of death (aHR=0.38, 95% CI:0.17-0.83, p=0.02) and/or mechanical ventilation (aHR=0.61, 95% CI:0.40-0.92; p=0.02) in patients who received tocilizumab over the standard of care [10]. On the contrary, initial randomized clinical trials, did not show any benefit [7,11], while recent large randomized trials suggest reduced likelihood of progression to IMV and death in patients with severe COVID-19 pneumonia treated with tocilizumab [12–14]. In particular, an open-label randomized multicenter study conducted in Italy including 126 patients with PaO2/FiO2 between 200 and 300 mmHg did not show tocilizumab benefit on disease progression [7]. On the other hand, REMAP-CAP trial showed improved 90-day survival in critically ill patients receiving organ support who were treated with tocilizumab or sarilumab (HR=1.61, 95% CI: 1.25-2.08) [12]. Moreover, RECOVERY trial reports that patients receiving tocilizumab had improved survival and were less likely to reach the composite endpoint of IMV or death (33% vs. 38%; risk ratio 0.85; 95% CI 0.78-0.93; p=0.0005) [14].

These conflicting results could be justified by diverse inclusion criteria, small sample size, competing effect of other treatments such as glucocorticoids, but also by different stages of the disease when tocilizumab is administered [15].”

METHODS

1) Please specify if the patients’ cohort here described overlaps with the one of the TESEO study. If so, the authors should clarify this.

Authors’ answer:

The following sentence has been changed:

“This study included consecutive adult patients (≥18 years) admitted to Infectious Disease Clinic of the University Hospital of Modena, Italy, from 21 February to 10 April 2020, with radiological findings suggestive for SARS-CoV-2 pneumonia and confirmed by PCR method on nasopharyngeal swab, belonging to the previously described TESEO cohort [11], but restricting the analyses to the Modena’s cohort subset only.”

The following sentence have been added:

“Inclusion and exclusion criteria were the same of the parental cohort [11].”

2) Please specify whether every admitted patient in the study period was enrolled and the eventual exclusion criteria.

Authors’ answer:

The following sentence have been added:

“Inclusion and exclusion criteria were the same of the parental cohort [11]. Patients that did not satisfy criteria for severe pneumonia were excluded from the analysis.”

3) Why did the authors not decide to use the complete WHO clinical progression scale as outcome?

Authors’ answer:

Considering that per inclusion criteria we enrolled only hospitalized patients, we could not use the complete WHO clinical progression scale which includes also non hospitalized patients.

The following sentence has been added in the methods:

“Additionally, as per inclusion criteria only hospitalized patients were enrolled, we were not able to use the complete scale proposed by WHO R&D Blueprint expert group, which also comprises non-hospitalized patients.”

4) “Two respiratory support states, such as non-invasive ventilation (NIV) or invasive mechanical ventilation (IMV) states were grouped to have a properly sized single category”. This is an important limitation of the study, as it is impossible to distinguish the escalation from NIV to IMV but also the de-escalation from IMV to NIV. The authors should address the limitation and, if possible, provide alternative analysis with the two distinct categories.

Authors’ answer:

We agree with the reviewer that this is a major limitation but was a necessary step to obtain sufficient statistical power. This has been acknowledged as the limitation of the study in the discussion:

“Fourthly, NIV and IMV were grouped together in order to have properly sized single category, but this choice did not permit to distinguish the escalation from NIV to IMV and de-escalation from IMV to NIV as important steps in management of severe COVID-19 pneumonia.”

5) In the covariates subparagraph, the authors should be express more clearly if glucocorticoids were considered potential confounders and the analysis was controlled for their administration.

Authors’ answer:

As stated in the notes of each multivariable adjusted model and in the description of the statistics, use of glucocorticoids was considered as a potential confounder. This has been highlighted in the description of statistical analysis:

“Age, gender, SOFA score, time from the onset of COVID-19 in Modena and other pharmacological interventions including glucocorticoids were also evaluated as potential effect modifiers by adding interaction terms in the multivariable multi-state model.”

Also, the following sentence has been added in the paragraph regarding covariates: 

“Use of glucocorticoids was considered as a potential confounder in adjusted models.”

DISCUSSION

1) The authors may consider providing brief mentioning current guidelines recommendation towards or against tocilizumab for covid19

Authors’ answer:

As suggested by the reviewer, this has been modified in the discussion:

“Moreover, encouraging results from large randomized clinical trials [12,14] could provide a greater level of evidence and recommendations in the current guidelines for the treatment of COVID-19 pneumonia.”

CONCLUSIONS

2) The authors conclude saying that “we were able to show the positive impact of tocilizumab used in different disease stages, especially in the early onset of ARDS”. I would change the term “ARDS” with “COVID-19”. Furthermore, I would rephrase the conclusions with a more conservative approach. Data are still conflicting and the nonrandomized study design suggest not to over rely on such results, while awaiting for further RCTs.

Authors’ answer:

The conclusion has been mitigated according to the reviewer’s suggestion:

“To conclude, we were able to show the positive impact of tocilizumab used in different disease stages depicted by respiratory support states. The use of the multi-state Markov model allowed to harmonize the heterogeneous mortality and recovery endpoints and summarize results with stack probability plots. This approach could inform randomized clinical trials regarding tocilizumab, support disease management and hospital decision making.”

Reviewer #2: In the paper by J. Milic, the authors analyze the results of tocilizumab therapy on a cohort of covid-19 affected patients, already described on a previous prospective observational study (Guaraldi G eta al 2020). In this second paper they use a different statistical approach: the Markov multy-state model. The tocilizumab group, compared with the control group, clearly show worst clinical conditions at the time of enrollment, with worst respiratory function, higher flogosis parameters and lower lymphocytes counts, nevertheless the tocilizumab group presents a better survival and in general a better recovery. For these reasons, in the opinion of the reviewer, the starting differences between the two groups do not impair the results.

As the authors asses in the discussion section, the scientific production on this field is in part controversial, in particular they refer to the paper by Salvarani C et al 2020. Have the authors any possible explanation or comments on these contrasting results?

Authors’ answer:

Conflicting results between the trials could be justified by different inclusion criteria, small sample size, administration of other treatments such as glucocorticoids, but also by the stage of the disease when tocilizumab is administered. We have updated the recent literature regarding tocilizumab in the background and discussion (please, see above).

In the result section the sentence about the lymphocytes counts in the two group seems to be inverted: 1.0 x 106/L should refer to tocilizumab group, as indicated in the table not to the control group as stated in the text.

Authors’ answer:

We thank the reviewer for pointing out this mistake that has been corrected in the text.

Best regards,

Dr Jovana Milic

Prof. Giovanni Guaraldi

---

## [Editor Report · Decision Letter 1]

26 Apr 2021

The impact of tocilizumab on respiratory support states transition and clinical outcomes in COVID-19 patients. A Markov model multi-state study

PONE-D-20-37536R1

Dear Dr. Milic,

We’re pleased to inform you that your manuscript has been judged scientifically suitable for publication and will be formally accepted for publication once it meets all outstanding technical requirements.

Kind regards,

Chiara Lazzeri

Academic Editor

PLOS ONE
---

## [Editor Report · Acceptance letter]

5 Aug 2021

PONE-D-20-37536R1 

The impact of tocilizumab on respiratory support states transition and clinical outcomes in COVID-19 patients. A Markov model multi-state study 

Dear Dr. Milic:

I'm pleased to inform you that your manuscript has been deemed suitable for publication in PLOS ONE. Congratulations! Your manuscript is now with our production department. 

Kind regards, 

on behalf of

Dr. Chiara Lazzeri 

Academic Editor

PLOS ONE